Stable isotope and fatty acid analyses reveal significant differences in trophic niches of smooth hammerhead Sphyrna zygaena (Carcharhiniformes) among three nursery areas in northern Humboldt Current System

Segura-Cobeña Eduardo esegura@magister.ucsc.cl 1 2
Alfaro-Shigueto Joanna 2 3
Mangel Jeffrey 2 4
Urzua Angel 5 6
Górski Konrad konrad.gorski@uach.cl 5 6 7
1 Programa de Magister en Ecología Marina, Facultad de Ciencias, Universidad Católica de la Santísima Concepción , Concepción , Chile
2 ProDelphinus , Lima , Perú
3 Carrera de Biología Marina, Facultad de Ciencias Veterinarias y Biológicas, Universidad Cientifica del Sur , Lima , Perú
4 Centre for Ecology and Conservation, School of Biosciences, Cornwall Campus, University of Exeter , Penryn , Cornwall , United Kingdom
5 Departamento de Ecología, Facultad de Ciencias, Universidad Católica de la Santísima Concepción , Concepción , Chile
6 Centro de Investigación en Biodiversidad y Ambientes Sustentables (CIBAS), Universidad Católica de la Santísima Concepción , Concepción , Chile
7 Instituto de Ciencias Marinas y Limnológicas, Facultad de Ciencias, Universidad Austral de Chile , Valdivia , Chile
de Rezende Carlos
Electronic publication date: 2021 Apr 22
Publication date: 2021
Volume: 9
Electronic Location ID: e11283
Received 2020 Dec 7; Accepted 2021 Mar 24
Copyright: ©2021 Segura-Cobeña et al.
Copyright year: 2021
Copyright holder: Segura-Cobeña et al.
License: This is an open access article distributed under the terms of the Creative Commons Attribution License, which permits unrestricted use, distribution, reproduction and adaptation in any medium and for any purpose provided that it is properly attributed. For attribution, the original author(s), title, publication source (PeerJ) and either DOI or URL of the article must be cited.
License URL: https://creativecommons.org/licenses/by/4.0/

Keywords: Humboldt Current System, Sharks nursery areas, Denitrification, Feeding dynamics, Isotopic niche

Funding: Fondo para la Innovación, la Ciencia y la Tecnología PIBA-369-2014 Centro de Investigaciones en Biodiversidad y Ambientes Sustentables (CIBAS) of the UCSC Small Grant SOSF 521 from the Save Our Seas Foundation This study was supported by Fondo para la Innovación, la Ciencia y la Tecnología (PIBA-369-2014) and Centro de Investigaciones en Biodiversidad y Ambientes Sustentables (CIBAS) of the UCSC. This study was performed with the support of Small Grant SOSF 521 from the Save Our Seas Foundation. The funders had no role in study design, data collection and analysis, decision to publish, or preparation of the manuscript.

==============================
Fishery pressure on nursery areas of smooth hammerhead in northern Peruvian coast have become a serious threat to sustainability of this resource. Even though, some management actions focused on conservation of the smooth hammerhead populations were proposed in recent years, their scientific foundations are often limited, and biomass of smooth hammerhead in Peruvian waters continues to decrease. To inform management and conservation, this study aims to evaluate the trophic niche of smooth hammerhead juveniles from three nursery areas in the northern Peruvian coast using stable isotope and fatty acid analyses. First, we compared the environmental characteristics of each nursery area (i.e., sea surface temperature and chlorophyll-a concentration) and concluded that nursery areas differed significantly and consistently in sea surface temperature. Subsequently, we evaluated isotopic composition of carbon and nitrogen and fatty acid profiles of muscle and liver tissues collected from juvenile smooth hammerhead from each nursery area. We found that juvenile smooth hammerhead captured in San José were enriched in heavier 13C and 15N isotopes compared to those captured in Máncora and Salaverry. Furthermore, the broadest isotopic niches were observed in juveniles from Máncora, whereas isotopic niches of juveniles from Salaverry and San José were narrower. This difference is primarily driven by the Humboldt Current System and associated upwelling of cold and nutrient rich water that drives increased primary production in San José and, to a less extent, in Salaverry. Compared to smooth hammerhead juveniles from Máncora, those from San José and Salaverry were characterised by higher essential fatty acid concentrations related to pelagic and migratory prey. We conclude that smooth hammerhead juveniles from three nursery areas in the northern Peruvian coast differ significantly in their trophic niches. Thus, management and conservation efforts should consider each nursery area as a unique juvenile stock associated with a unique ecosystem and recognize the dependence of smooth hammerhead recruitment in San José and Salaverry on the productivity driven by the Humboldt Current System.

Introduction

Many shark species are globally threatened by fisheries due to a significant demand for shark fins in some Asian countries (e.g., China, Korea, Vietnam) (Jacquet et al., 2008; Frisch et al., 2016). Furthermore, due to some life history characteristics such as late sexual maturity, low fecundity and growth rates, high longevity and long gestation periods many shark populations are highly vulnerable (Cortés, 2000). Humboldt Current System (HCS) makes northern Peru fisheries of a variety of species among the most productive on Earth (Kämpf & Chapman, 2016). As a result, Peru ranks among 20 most important shark fin exporters in the world (Dent & Clarke, 2015).

The smooth hammerhead Sphyrna zygaena (Linnaeus, 1758) is a species of international concern, categorised as vulnerable by the International Union for Conservation of Nature (IUCN) and added to Appendix II of the Convention on International Trade in Endangered Species of Wild Fauna and Flora CITES (Rigby et al., 2019). Nevertheless, the smooth hammerhead is among the most appreciated shark species on the Asian market and has become the third most frequently captured shark species in Peru summing up to 15% of the total shark landings (Clarke et al., 2006; González-Pestana, Kouri & Velez-zuazo, 2016). Furthermore, more than half of these landings correspond to artisanal fishers from the central and northern coast of Peru that operate in three smooth hammerhead nursery areas (González-Pestana, Kouri & Velez-zuazo, 2016).

In the context of nature conservation, a nursery area is a zone where neonates and juveniles of a species are in high abundance and, as such, it is expected to offer some benefits to the juvenile population such as increased food availability or protection from predators (Simpfendorfer & Milward, 1993; Beck et al., 2001; Heupel, Carlson & Simpfendorfer, 2007). Globally, many nursery areas have been described for the smooth hammerhead e.g., in the Atlantic ocean on the coast of Uruguay, on the Brazilian continental shelf and on the coast of South Africa; in the Pacific ocean on the occidental shelf of the North Island at New Zealand (Smale, 1991; Vooren, Klippel & Galina, 2005; Doño, 2008; Francis, 2016).

Three nursery areas of smooth hammerhead have been described in the HCS based on the presence and abundance of neonates and juveniles (total length between 53 and 150 cm) (González-Pestana, 2014; González-Pestana, 2018). One is located in the Tropical East Pacific Marine Province (TEP-MP) at approximately 4°S in proximity of Máncora Port and is characterized by high fish diversity (Spalding et al., 2007; Ibanez-Erquiaga et al., 2018). The other two nursery areas are located in the Warm Temperate Southeastern Pacific Marina Province (WTSP-MP), one at approximately 6°40′S in proximity of San José Port and one at approximately at 8°15′S in proximity of Salaverry Port (Spalding et al., 2007). Both of these nursery areas located in WTSP-MP are characterized by high productivity driven by the HCS and high fish biomass and diversity (Chavez et al., 2008; Ibanez-Erquiaga et al., 2018). Furthermore, the nursery area at San José Port is characterised by the presence of small islands and is limited to continental shelf, whereas the nursery area at Salaverry Port does not accommodate islands and spreads further out to oceanic waters.

The ecological niche is commonly defined as a combination of environmental conditions, biotic and abiotic variables, in which the species can persist, use resources, and impact its environment (McGill et al., 2006). The trophic niche is a part of the ecological niche and it describes all trophic interactions of a population or species, its food resources and feeding area (Potapov, Tiunov & Scheu, 2018; Shipley et al., 2018). Since the biochemical composition of an organism is directly related to its feeding habits, it is possible to evaluate an approximation of the trophic niche of a species through analyses of its stable isotopes and fatty acids (Bec et al., 2011). Specifically, the isotopic composition of an organism, its isotopic niche, depends on its diet, feeding habits and trophic interactions (Newsome et al., 2007). Fatty acid composition can also be used as a trophic marker because it directly relates to the lipid reserve obtained by feeding (Dalsgaard et al., 2003). Even though the majority of fatty acids can be synthetized by an organism, still most of them are obtained directly from consumed food (Iverson et al., 2004). As such, a combination of stable isotope analyses and evaluation of fatty acid profiles can be a powerful tool to obtain comprehensive, time integrated (weeks to a year) assessment of diet and feeding behaviour and, therefore, approximate the trophic niche of a marine predator (Hooker et al., 2001).

Nursery areas are expected to bring benefits to juvenile populations and are essential for adult population recruitment (Heupel, Carlson & Simpfendorfer, 2007). Increasing fishery pressure on nurseries of the smooth hammerhead in northern Peruvian waters can be a serious threat to sustainability of this fishery. However, ecosystem-based fisheries management has not been implemented in Peruvian waters due to lack of data on biology and ecology of commercially important elasmobranch species. This study aims to evaluate the trophic niche of juvenile smooth hammerheads from three nursery areas using stable isotope and fatty acid analysis in northern Peruvian waters to inform conservation. As each of the nursery area is under influence of different water masses and is characterised by different environmental conditions, we expect trophic niches of smooth hammerhead juveniles to be significantly different among nursery areas.

Methodology

Study area

Data on presence of juvenile smooth hammerheads recorded by artisanal fishers and monitors of ProDelphinus, an non-governmental organization (NGO) dedicated to fishery conservation, between June 2014 and December 2018 were used to delimitate specific nursery areas in northern Humboldt Current System (HCS) (Fig. 1). Sea surface temperature (SST) and chlorophyll-a (Chl-a) data series were obtained from NASA MODIS-Aqua (Moderate Resolution Imaging Spectrometer Aqua) via Oceancolor Data Downloader (https://oceandata.sci.gsfc.nasa.gov/MODIS-Aqua/). Satellite images with 4 km2 spatial resolution were downloaded at seasonal scale between 2010 and 2019. Specific seasons were divided as follows: Summer (December–February), Autumn (March–May), Winter (June–August) and Spring (September–November). In addition, information on occurrence of ENSO oscillation events was obtained from NOAA Climate Prediction Center (https://www.cpc.ncep.noaa.gov/).

Figure 1 Landing ports (dots) and nursery areas smooth hammerhead Sphyrna zygaena (polygons).

Areas were delimitated based on points of presence of juveniles Smooth Hammerhead from catches by artisanal fishery. Data source: ProDelphinus. Continental shelf area is indicated in light grey.

Sample collection

Juvenile smooth hammerheads that were captured as bycatch by artisanal gillnet fishers between February and March 2019 in each nursery area were used with the approval of Peruvian Ministerio de la Producción (PRODUCE), registry N∘ 00008103-2019. Muscle and liver tissue samples were extracted from each individual and its total length (TL) was registered. Due to time limitations some fishers provided sample of only one of the tissues. Since smooth hammerheads are born with a large liver that carries nutrients from maternal heritage that are expected to affect isotopic signature and fatty acid profiles of neonates (Francis, 1994; Olin et al., 2011), only juveniles between 70 and 120 cm TL were considered for this study. Furthermore, muscle tissue of sympatric fish and squid species that are usually caught together with smooth hammerhead were also collected by artisanal fishers in the same period (the number of available samples depended on both availability of fish and squid and time that could be dedicated by the fishers for sample collection). Tissues samples of smooth hammerheads and sympatric species were preserved in 80% ethanol (since collection areas were remote and no others preservation options were accessible), and upon arrival to the laboratory were lyophilized for further analyses. Ethanol preservation may either deplete or enrich isotopic signatures of carbon and nitrogen, however there is no consensus whether a correction factor should be applied or not (Sarakinos, Johnson & Vander Zanden, 2002; Xu et al., 2011). Direct evaluations of the effects of ethanol fixation on fatty acid profiles are scarce, but Phleger et al. (2001) found no differences in fatty acid profile of ethanol-preserved samples of rock lobster in comparison with frozen or lyophilized samples. Furthermore, all samples in our study were preserved with the same method allowing for valid comparisons among the nursery areas. The total of smooth hammerhead muscle samples obtained was 10 in Máncora (115.2 ± 5.7 cm TL), 19 in San José (80.2 ± 6.3 cm TL) and 14 in Salaverry (81.1 ± 2.5 cm TL). The total smooth hammerhead liver samples obtained was 13 in Máncora (114.4 ± 5.2 cm TL), 17 in San José (80.3 ± 6.6 cm TL) and 14 in Salaverry (81.1 ± 2.5 cm TL).

Stable isotope analysis

Lipid content of tissue subsamples that were used for stable isotope analyses was extracted to prevent the alteration of carbon stable isotope signatures (Sweeting, Polunin & Jennings, 2006). Lipid extraction was performed following the methods described by Folch, Lees & Sloane (1957) modified by Cequier-Sánchez et al. (2008) and Urzúa & Anger (2013). We used 20 mg of tissue and 5 ml of the solvent dichloromethane: methanol (2:1). Subsequently, all muscle samples were dried, pulverized and 1 mg of each sample was placed inside a tin capsule for analyses of the composition of carbon and nitrogen stable isotopes. Samples were sent to stable isotopes laboratory at University of California in Davis, USA (https://stableisotopefacility.ucdavis.edu). To assess the variation in isotopic ratios of carbon and nitrogen δ notations were calculated following the equation (Coplen, 2011): δhX=RsampleRstandard−1

Where X is the element, h is the high mass number, Rsample is the high mass-to-low mass isotope ratio of the sample and Rstandard is the high mass-to-low mass isotope ratio of the standard (Vienna Pee Dee Belemnite for carbon and atmospheric nitrogen for nitrogen). The δ ratio was expressed in parts per thousand (‰).

When working with stable isotope in elasmobranch tissue, it is recommended to extract lipid and urea (Ingram et al., 2007; Li et al., 2016; Carlisle et al., 2017). In this study, as urea extractions were not performed, we used the correction factor from Li et al. (2016) for smooth hammerhead to correct the δ15N values following the equation: δ15NLE+UE=0.984∗δ15NLE+2.063

Where δ15NLE+UE is the value of δ15N corrected with lipid extraction (LE) and urea extraction (UE).

Fatty acid analysis

The fatty acid profile was determined following the methods presented by Urzúa & Anger (2011). Fatty acid methyl esters (FAMEs) were measured after preparation using the total lipid extracted from each sample (Couturier et al., 2020). Total lipid extracts were esterified using methanolic sulphur acid at 70 °C for 1 h in a Thermo-Shaker (MRC model DBS-001). Subsequently, fatty acids were rinsed using 6 ml of n-hexane. Finally, the measurement of FAMEs was performed using a gas chromatograph (Agilent, model 7890A) equipped with a DB-225 column (J & W Scientific, 30 m in length, 0.25 mm inner diameter and 0.25 µm film thickness) at a range of temperatures. Individual FAMEs were identified by comparison to known fatty acid standards of marine origin using chromatograph software (Agilent ChemStation, USA) and certificate material Supelco 37 FAME mix 47885-U, and quantified by means of the response factor to internal standard 23:0 FA added prior to transmethylation (Malzahn et al., 2007; Urzúa & Anger, 2011).

Statistical analyses

To compare the environmental variables among three nursery areas we used repeated measures ANOVA (ANOVA-RM) and post-hoc test with Bonferroni correction. The factors used in the model were: nursery areas as groups, years as intra-case factor and seasons as cases. PERMANOVA was used to compare stable isotope signatures (δ13C and δ15N) among nursery areas (Máncora, San José, Salaverry) and tissue types (muscle, liver) (Anderson, 2001; Mcardle et al., 2001). Subsequently, niche sizes and niche overlaps of the smooth hammerheads were compared among the nursery areas using nicheROVER package in R (Lysy, Stasko & Swanson, 2014; Swanson et al., 2015). This package evaluates the niche size (NS) as the 95% of the region occupied by the species or population. Niche overlap is calculated by a Bayesian framework with 95% of the area of the ellipses based on data points. Subsequently, the overlap is calculated as the probability of the individual from one area to share the isotopic space with an individual from another area. An overlap higher than 60% is consider significant by criteria used in niche studies (Schoener, 1968). In addition, the package “tRophicPosition” was used to compare trophic position (TP) of juvenile smooth hammerheads among nursery areas. This package calculates Bayesian TP estimates using δ15N from the base line (sympatric species) (Quezada-Romegialli et al., 2018). We used discrimination factors specific for smooth hammerhead that were estimated by Kim et al. (2012): 1.7 ±  0.5 for δ13C and 3.7 ± 0.4 for δ15N. Sympatric species were assumed to represent TP of 3.6, which is the mean value of the trophic position of secondary fish consumers in the HCS (Espinoza et al., 2017). Comparisons of isotopic signatures between smooth hammerhead juveniles and sympatric species were based on liver signatures of smooth hammerheads because liver has a higher isotopic turnover rate compared to muscle and therefore it is expected to reflect more recent diet and be less affected by maternal signature.

To compare fatty acid diversity among nursery areas, ANOVA was performed on the Shannon diversity index (H’) calculated for fatty acid profiles in each tissue and nursery area (Shannon, 1948). Data were log transformed prior to analyses to meet ANOVA assumptions. Subsequently, PERMANOVA based on Bray-Curtis dissimilarity matrix was used to compare fatty acid profiles and isotopic signatures (δ13C and δ15N) together among nursery areas (estimated probabilities were based on 999 permutations). Absolute values of δ13C were used and all data were square root transformed prior to analyses to reduce the effects of outliers. Subsequently, to assess which response variables were the most important drivers of differences among nursery areas we used Principal Coordinate Ordination (PCO) (Anderson, 2017). Variables with Pearson correlation higher than 0.5 were plotted (Meyer et al., 2019). All multivariate statistical analyses were executed in PRIMER 6 (Plymouth Routines In Multivariate Ecological Research) (Clarke & Warwick, 2001).

Results

Environmental factors

We found significant differences in SST (ANOVA-RM of Area * Years: F(16,3) = 2.26; p = 0.02) and chlorophyll a (ANOVA-RM of Area * Years: F(16,3) = 4.78; p < 0.001) among the nursery areas. Specifically, Máncora was characterized by SST consistently 2 °C higher compared to Salaverry. Furthermore, SST in Salaverry was consistently 1 °C higher compared to San José (Fig. S1). In contrast, chlorophyll a concentration in San José were consistently about 2 mg m−3 higher compared to Máncora and Salaverry (Fig. S2).

Differences in isotopic niches among nursery areas

Smooth hammerhead juveniles captured in Salaverry had similar isotopic signatures compared to those captured in Máncora but were slightly enriched in 15N (higher δ15N values; Fig. 2). Furthermore, the highest variability of isotopic signatures in both tissues was observed in smooth hammerhead juveniles captured in Máncora. Smooth hammerhead juveniles captured in San José were characterised by isotopic signatures enriched in 13C and 15N in both liver and muscle tissues (δ13C and δ15N values significantly higher) compared to those captured in Salaverry and Máncora and these differences were more marked in liver tissue (Fig. 2; PERMANOVA; F = 39; p = 0.001). Elliptical isotopic niche projections showed significant overlap between the three nursery areas (>60%; Fig. 3; Table 1). The broadest isotopic niches were observed in smooth hammerhead juveniles from Máncora, whereas isotopic niches of smooth hammerhead juveniles from Salaverry and San José were narrower. These differences were consistent between results based on muscle and liver tissues. Sympatric species captured in all nursery areas were characterized by carbon signatures consistently enriched in 13C compared to carbon signatures of smooth hammerhead juveniles from the same areas (Fig. 4).

Figure 2 Mean and standard deviation of the δ13C and δ15N of smooth hammerhead Sphyrna zygaena by nursery area and tissue type.

Results for muscle tissue are represented by squares and results for liver tissue are represented by triangles. Máncora, red; San José, yellow; Salaverry, blue.

Figure 3 Four elliptical projections of the niche region (NR) of smooth hammerhead Sphyrna zygaena muscle (A) and liver (B) tissue from each nursery area.

Máncora, red; San José, yellow; Salaverry, blue.

Table 1 Mean niche size (NS) and niche overlap based on δ13C and δ15N of muscle and liver tissues of smooth hammerhead Sphyrna zygaena from the three nursery areas.

		Máncora	San José	Salaverry	NS	
Muscle	Máncora	–	71.39	74.31	5.23	
San José	69.92	–	87.27	3.75	
Salaverry	62.37	84.26	–	3.43	
Liver	Máncora	–	67.35	75.02	12.57	
San José	70.28	–	88.91	4.82	
Salaverry	64.71	82.91	–	7.13	

Figure 4 Mean and standard deviation of the δ13C and δ15N of sympatric species and scatterplot of the liver tissue δ13C and δ15N of the smooth hammerhead Sphyrna zygaena from each nursery area.

(A) Máncora; (B) San José; C: Salaverry; Eastern Pacific bonito (Sarda chiliensis); Spinetail mobula (Mobula japanica); Jumbo squid (Dosidicus gigas); Anchoveta (Engraulis ringens); South Pacific hake (Merluccius gayi); Minor stardrum (Stellifer minor); Drab tonguefish (Symphurus melanurus); Peruvian weakfish (Cynoscion analis); Suco croaker (Paralonchurus dumerilii); Grey mullet (Mugil cephalus); Pacific menhaden (Ethmidium maculatum); Peruvian morwong (Cheilodactylus variegatus); Point-Tuza croaker (Ophioscion scierus); Paloma pompano (Trachinotus paitensis); Longnose anchovy (Anchoa nasus).

Differences in fatty acid concentrations and diversity among nursery areas

Juvenile smooth hammerheads from Máncora were characterised by the lowest diversity of fatty acids in both muscle and liver tissues. Furthermore, only saturated fatty acids were recorded in muscle, while in the liver tissue saturated fatty acids were found in the highest concentrations (Table 2). Juvenile smooth hammerheads from Salaverry and San José were also characterised by the highest concentrations of saturated fatty acids, but polyunsaturated fatty acids were also found in these individuals in both muscle and liver tissues. The highest diversity (H’) of fatty acids in muscle tissue was found in smooth hammerhead juveniles from Salaverry (1.02 ± 0.6) followed by San José (1 ± 0.4), whereas diversity of fatty acids was significantly lower in smooth hammerhead juveniles from Máncora (0.46 ± 0.3) (F = 6.42; P = 0.004). Similar pattern was observed in diversity of fatty acids in liver tissue. Specifically, H’ of fatty acids of smooth hammerhead juveniles from Salaverry (1.93 ± 0.2) and San José (1.92 ± 0.1) were significantly higher compared to smooth hammerhead juveniles from Máncora (0.76 ± 0.2) (F = 6.42; P = 0.004).

Table 2 Muscle and liver fatty acid concentrations and Shannon fatty acid diversity of the smooth hammerhead Sphyrna zygaena in each nursery area.

	Muscle	Liver	
	Máncora	San José	Salaverry	Máncora	San José	Salaverry	
C11:0	nd	0.03 ±0.1	0.05 ±0.2	0.05 ±0.2	0.04 ±0.2	nd	
C12:0	nd	nd	nd	0.06 ±0.2	0.08 ±0.2	nd	
C13:0	nd	nd	nd	0.04 ±0.2	0.05 ±0.2	nd	
C14:0	nd	0.11 ±0.3	0.31 ±0.4	7.77 ±2.1a	12.86 ±4.7b	14.08 ±4.6b	
C15:0	nd	nd	nd	1.9 ±0.5a	2.25 ±0.7a,b	2.14 ±0.4b	
C16:0	1.35 ±0.2a	2.40 ±0.9b	2.22 ±1.0b	82.3 ±18.8a	88.6 ±25.9b	91.36 ±22.2b	
C17: 0	nd	nd	nd	4.11 ±0.8a	4.14 ±1a	5.71 ±0.1b	
C18:0	0.60 ±0.4a	1.59 ±0.7b	1.23 ±0.9a,b	23.9 ±2.5a	27.67 ±5.5a,b	29.6 ±5.1b	
C20:0	nd	nd	nd	0.24 ±0.5	0.14 ±0.4	0.18 ±0.6	
C22:0	nd	nd	0.33 ±0.9	nd	0.39 ±1.6	1.32 ±3.7	
C23:0	nd	nd	nd	0.69 ±1.7	2.01 ±3.8	nd	
C24:0	nd	nd	nd	nd	0.08 ±0.3	0.08 ±0.3	
TOTAL SFA	1.95 ±0.6a	4.12 ±1.7b	4.14 ±2.2b	121.1 ±22.5	138.3 ±37.5	143.9 ±29.3	
C14:1	nd	nd	nd	0.13 ±0.5	0.05 ±0.2	nd	
C16:1	nd	0.09 ±0.3	0.28 ±0.6	17.7 ±14.1	27 ±13.3	23.54 ±7.6	
C17:1	nd	nd	nd	0.48 ±1	0.68 ±1.4	nd	
C18:1n9	nd	1.03 ±1.1	0.98 ±1.1	44.63 ±19.4	41.18 ±14.4	48.81 ±20.5	
C20:1	nd	nd	nd	5.35 ±4.9a	3.39 ±3.2b	7.97 ±5.6a,b	
C22:1n9	nd	nd	nd	0.26 ±0.7	nd	0.26 ±0.8	
C24:1	nd	nd	nd	1.11 ±1.3	1.46 ±1.8	0.23 ±1	
TOTAL MUFA	nd	1.12 ±1.3	1.26 ±1.5	69.7 ±33.6	73.8 ±29	80.8 ±25.1	
C18:3n3	nd	nd	nd	0.94 ±1.5	1.09 ±1.5	1.1 ±1.5	
C20:3n3	nd	nd	nd	1.57 ±3.4	2.83 ±3.5	2.07 ±3.5	
C20:5n3	nd	0.08 ±0.3	nd	11.32 ±18.9	20.97 ±25.1	33.74 ±25.6	
C22:6n3	nd	0.94 ±1.8	1.41 ±1.8	11.07 ±14.2a	52.12 ±41.2b	42.99 ±43.5a,b	
TOTAL PUFAn3	nd	1.02 ±2.1	1.41 ±1.8	24.9 ±35.4a	77 ±63.4b	79.9 ±61.1b	
C18:2n6c	nd	nd	nd	nd	0.28 ±0.8	nd	
C18:2n6t	nd	nd	nd	1.21 ±1.9a	2.1 ±2.1a,b	3.55 ±2.3b	
C18:3n6	nd	nd	nd	0.2 ±0.7	0.62 ±1.1	nd	
C20:3n6	nd	nd	nd	nd	0.05 ±0.2	nd	
TOTAL PUFAn6	nd	nd	nd	1.4 ±2.5a	3.1 ±3.5b	3.5 ±2.3b	
C20:2	nd	nd	nd	0.2 ±0.7	0.26 ±0.7	nd	
TOTAL PUFA	nd	1.02 ±2.1	1.41 ±1.8	26.5 ±38	80.3 ±64.1	83.4 ±62.2	
TOTAL FAs	1.95 ±0.6a	6.26 ±4.5b	6.82 ±4.6b	217.2 ±70.1a	292.4 ±98.8b	308.1 ±90.4b	
Shannon’s Index	0.46 ±0.32a	0.99 ±0.43b	1.05 ±0.58b	1.76 ±0.21a	1.93 ±0.19b	1.92 ±0.1b	
Notes.

FAs not detected are indicated with “nd”. Superscripted, lowercase letters indicate significant differences between seasons and locations (two-way ANOVA, P < 0.05).

Mean ± standard deviation in mg g PS−1.

SFA saturated fatty acids

MUFA monounsaturated fatty acids

PUFA polyunsaturated fatty acids

TFA total fatty acids

Differences in trophic niche based on isotopic signatures and fatty acid profiles among the nursery areas

PERMANOVAs based on fatty acid concentrations together with isotopic signatures (δ13C and δ15N) indicated significant differences between smooth hammerhead juveniles from Máncora and smooth hammerhead juveniles from San José and Salaverry in muscle (F = 4.83; P = 0.004) and liver (F = 3.54; P = 0.002). Furthermore, PCO based on fatty acid concentrations together with isotopic signatures in muscle tissue clearly separates smooth hammerhead juveniles captured in San José and Salaverry from those captured in Máncora mainly due to higher overall fatty acid concentrations of those captured in San José and Salaverry (Fig. 5A). Similar pattern was observed in PCO based on both fatty acid concentrations and isotopic signatures in liver tissue but here, in addition to higher fatty acid concentration, smooth hammerhead juveniles from Salaverry and San José were also characterized by carbon enriched in heavier 13C isotope (Fig. 5B).

Figure 5 Principal Coordinate Ordination (PCO) based on δ13C and δ15N and fatty acid profiles of muscle (A) and liver (B) tissue of smooth hammerhead Sphyrna zygaena from each nursery area.

Máncora (red), San José (yellow) and Salaverry (blue). Vectors show variables with Pearson correlation ≥ 0.5.

Discussion

Significant differences in trophic niches based on stable isotope signatures and fatty acid profiles among three nursery areas suggest distinct trophic dynamics of juvenile smooth hammerheads in each area largely driven by the HCS. Furthermore, isotopic signatures corroborate that juvenile smooth hammerheads from San José nursery area feed on prey enriched in heavier isotopes of carbon and nitrogen, compared to juvenile smooth hammerheads from Salaverry and Máncora nursery areas. In addition, smooth hammerheads from the San José and Salaverry nursery areas were characterised by similar fatty acid profiles that were significantly more diverse compared to the fatty acid profile of smooth hammerheads from Máncora.

Isotopic niche variation among the nursery areas

The influence of the HCS and related coastal upwelling amplified by the extensive continental shelf results in low water temperatures and high nutrient concentrations in the nursery areas of San José and Salaverry (Chavez et al., 2008; Morales et al., 2019). The differences we observed in temperature and chlorophyll-a among the areas are maintained over time despite the seasonal variability related to El Niño Southern Oscillation (ENSO) cycles (Fiedler, 2002). Carbon signatures are also directly affected by the influence of HCS that result in enrichment in the heavier 13C isotope between 7 and 15°S (Echevin et al., 2008; Argüelles et al., 2012). Furthermore, the presence of an Oxygen Minimum Zone (OMZ) in the northern HCS intensifies denitrification and enrichment of 15N isotope available for photosynthesis (Liu & Kaplan, 1989; Chavez et al., 2008; Paulmier & Ruiz-Pino, 2009). This influence of the HCS can explain enrichment of juvenile smooth hammerheads captured in the San José nursery area in 13C and 15N isotopes. Similar enrichment could be expected for juvenile smooth hammerhead captured in Salaverry nursery area. However, juvenile smooth hammerheads from Salaverry were characterised by isotopic signatures and isotopic niche width similar to those captured in Máncora with lower concentrations of heavier 13C and 15N isotopes. This is probably because juvenile smooth hammerheads from Salaverry feed in more oceanic waters, where carbon and nitrogen isotopic signatures are characterized by lower concentrations of 13C and 15N isotopes, similar to those observed in Máncora nursery area (Echevin et al., 2008; Argüelles et al., 2012; Rabehagasoa et al., 2012). The probability that juvenile smooth hammerheads captured in Salaverry and Máncora migrate between these two nursery areas is low since smooth hammerheads are characterized by high fidelity to nursery areas where they were born (Holland et al., 1993).

Sympatric species, mostly fishes, that were evaluated within this study showed isotopic signatures enriched in 13C isotope compared to isotopic signature in the livers of juvenile smooth hammerheads. Therefore, species that were available for this study most probably are not consumed by juvenile smooth hammerheads. Indeed, squids, species from oceanic water, were previously documented to be the most important prey of juvenile hammerheads based on stomach content analyses (Bornatowski et al., 2007; Galván-magaña et al., 2013; González-Pestana et al., 2017; Dicken et al., 2018). Few jumbo squid Dosidicus gigas samples that were analyzed within this study were characterized by lower concentrations of 15N isotope compared to smooth hammerhead juveniles. This suggests jumbo squid as potential prey of smooth hammerhead, however analyses of larger amount of samples is needed to corroborate this finding. Other recent studies based on stable isotope analyses suggest that also demersal fishes and crabs may significantly contribute to the diet of juvenile smooth hammerheads (Kiszka et al., 2015; Loor-andrade et al., 2015; Rosende-Pereiro et al., 2020). Exhaustive sampling of sympatric species and development of isotopic baselines in all nursery areas including at least two seasons is recommended to elucidate the specific prey contributing to the diet of juvenile smooth hammerheads.

Fatty acid profiles variation among nursery areas

Fatty acid biomarkers complement analyses of stable isotopes and allow to elucidate further differences in trophic niches among nursery areas. The presence of polyunsaturated and saturated fatty acids (PUFA and SFA) in muscle and liver are indicators of pelagic zooplanktivorous prey such as squids and fishes (Pethybridge et al., 2010; Rohner et al., 2013). Juvenile smooth hammerheads from San José and Salaverry were characterized by higher concentrations of these fatty acids, probably due to a higher biomass of pelagic prey in these nursery areas driven by the HCS. The meristic acid (C14:0) detected in muscle and liver tissues of smooth hammerhead juveniles from San José and Salaverry nursery areas may indicate higher abundances of proteobacteria and diatoms in these areas most probably as a result of the HCS and coastal upwelling (Dalsgaard et al., 2003). Polyunsaturated EPA (C20:5n3) and DHA (C22:6n3) are the most relevant as dietary indicators as they cannot be synthetized by sharks (Turner & Rooker, 2005). These fatty acids are positively correlated to total length as they are usually depleted in sharks under one year (Wai et al., 2011; Wai et al., 2012; Belicka et al., 2012). Our results do not corroborate this relationship for smooth hammerhead juveniles as the largest individuals were captured in Máncora and were characterised by the lowest concentrations of EPA and DHA. In contrast, smooth hammerhead juveniles from San José and Salaverry nursery areas were characterised by higher concentration of both EPA and DHA perhaps due to lower temperatures and higher abundances of prey such as squids or migratory fishes driven by the influence of the HCS (Bell & Sargent, 1986; Saito, Ishihara & Murase, 1997; Semeniuk, Speers-Roesch & Rothley, 2007; Beckmann et al., 2013).

In comparison with previous studies, we found lower diversity of fatty acids in muscle tissue of smooth hammerheads (Davidson et al., 2011; Davidson et al., 2014). This difference could be related to the differences in extraction protocol as studies by Davidson et al. used 20 mg of lipids, while due to limited sample availability we used the lipids extracted from 20 mg of tissue. We still found the results of muscle tissue analyses worth presenting as the fatty acid profiles registered in liver tissues were similar to those registered in muscle tissue of the same individuals and, as expected, the concentrations in liver tissue were higher.

Smooth hammerhead trophic niche and its implications for fishery management and conservation

Integration of the stable isotope signatures and fatty acid profiles allowed differentiation of smooth hammerhead juveniles among San José, Salaverry and Máncora nursery areas. The trophic niche of a species or population consists of biotic and abiotic variables related to feeding habits and here we document that smooth hammerhead juveniles from each nursery area in Peruvian waters have distinct trophic niches characterised by specific types and quantities of prey they consume and their feeding habitats (oceanic or coastal waters).

The smooth hammerhead is among the elasmobranchs species most frequently landed in the Peru and its catches severely declined during recent years (González-Pestana, Kouri & Velez-zuazo, 2016). Conservation initiatives and implementation of ecosystem based fishery management are urgently needed to sustain this fishery, however they lack essential baseline data (Lack et al., 2014). We document that smooth hammerhead juveniles from three nursery areas in northern Peru differ significantly in their diets and trophic niches. Thus, management and conservation efforts should consider each nursery area as a unique juvenile stock associated with unique ecosystem and recognize the dependence of smooth hammerhead recruitment in San José and Salaverry on the productivity driven by the Humboldt Current System. Additional research could be undertaken to further refine our understanding of the spatial, temporal and environmental characteristics of these nursery areas and their stability over time given potential ENSO impacts. Our results and future studies may inform ecosystem-based fishery management that takes into account the entire ecosystem rather than a single species (Kinney & Simpfendorfer, 2009; Mason et al., 2020). New measures could build upon existing seasonal bans and landings restriction to take into account protection of distinct nursery areas. To be most effective, any future monitoring or management actions should involve fishers and communities that operate in these nursery areas to help design management measures that allow for both sustainable shark populations and sustainable fisheries. Ecosystem-based co-management may be instrumental in enhancing rapidly declining smooth hammerhead populations.

Supplemental Information

Supplemental Information 1 Mean and standard deviation of the Superficial Sea Temperature (SST ° C) seasonal between 2010–2018 of the nursery areas of Máncora (red), San José (yellow) and Salaverry (blue)

A, Autumn; W, Winter; Sp, Spring; and Su, Summer. Red background represents El Niño event and blue background represents La Niña event.

Click here for additional data file.

Supplemental Information 2 Mean and standard deviation of Chlorophyll a concentration (Chl-a mg m −3) 2010 and 2018 in nursery areas of Máncora (red), San José (yellow) and Salaverry (blue)

A: Autumn; W: Winter; Sp: Spring; and Su: Summer. Red background represents El Niño event and blue background represents La Niña event.

Click here for additional data file.

Supplemental Information 3 PERMANOVA results of δ13C and δ15N of smooth hammerhead Sphyrna zygaena comparison among nursery areas and tissues

Click here for additional data file.

Supplemental Information 4 PERMANOVA results of δ13C and δ15N and fatty acid profile of smooth hammerhead Sphyrna zygaena muscle tissue comparison among nursery areas

Click here for additional data file.

Supplemental Information 5 PERMANOVA results of δ13C and δ15N and fatty acid profile of of smooth hammerhead Sphyrna zygaena liver tissue comparison among nursery areas

Click here for additional data file.

Supplemental Information 6 Isotopic sign and fatty acid profile of Smooth Hammerhead and other fishes

Click here for additional data file.

Artisanal fishers from Máncora, San José and Salaverry ports are greatly acknowledged for their help in sample collection. Colleagues from Ko_Lab and Laboratorio de Recursos Hidrobiológicos are acknowledged for help in sample processing.

Additional Information and Declarations

Competing Interests

Author Contributions

Field Study Permissions

Data Availability

The authors declare there are no competing interests. Eduardo Segura-Cobeña and Jeffrey Mangel are employees of ProDelphinus. Joanna Alfaro-Shigeto is a director of ProDelphinus.

Eduardo Segura-Cobeña conceived and designed the experiments, performed the experiments, analyzed the data, prepared figures and/or tables, authored or reviewed drafts of the paper, and approved the final draft.

Joanna Alfaro-Shigueto and Jeffrey Mangel performed the experiments, authored or reviewed drafts of the paper, and approved the final draft.

Angel Urzua conceived and designed the experiments, authored or reviewed drafts of the paper, and approved the final draft.

Konrad Górski conceived and designed the experiments, analyzed the data, authored or reviewed drafts of the paper, and approved the final draft.

The following information was supplied relating to field study approvals (i.e., approving body and any reference numbers):

Field sampling were approved by Dirección de Extracción para Consumo Humano Direct e Indirecto from Ministerio de la Producción (PRODUCE), Peru, Registro N° 00008103-2019; Resolución Directoral N° 429-2019-PRODUCE/DGPCHDI.

The following information was supplied regarding data availability:

Isotopic sign and fatty acid data are available in the Supplementary Files.

The shapefiles of the Smooth Hammerhead nursery areas, used for the environmental data extraction, are available at Zenodo: Eduardo Segura-Cobeña, Joanna Alfaro-Shigueto, Jeffrey Mangel, Angel Urzua, & Konrad Górski. (2020). Smooth Hammerhead peruvian nursery areas shapefiles. Zenodo. Dataset. http://doi.org/10.5281/zenodo.4300913.

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
