# Peer review of "Stable isotope and fatty acid analyses reveal significant differences in trophic niches of smooth hammerhead Sphyrna zygaena (Carcharhiniformes) among three nursery areas in northern Humboldt Current System"

_PeerJ, doi:10.7717/peerj.11283_

## Round 0.1 · original submission · Major Revisions

The article needs to make several changes, including methodological procedure for sample preservation with ethanol.

Reviewer 1 ·

Basic reporting

There are many figures and tables, and I suggest some reorganization in General comments.

Experimental design

The methodology to fix the samples prior to stable isotopes and fatty acids analyses is not clear/justify, as indicated in the General comments.

Validity of the findings

The conclusion should explore more the applicability of the results in local fishery management, as indicated in the General comments.

Additional comments

The manuscript adds important information that can be applied to fisheries management in Peru in relation to an endangered shark species. I have some comments/suggestions to be considered by the authors.
Figures and tables: There are many figures and tables. Join the table 1 and 2 in a single table; join fig. 6 and 7 in a single fig. (a and b, for instance); fig. 2 can be excluded because numerical data were already included in the main text.
Abstract:
Lines 30-36: The abstract 'introduction' is too long. Only one or two phrases are enough here.
Introduction:
lines 64-65: 'nowadays' with a reference of 2005? 15 years ago? Update the reference to support the information.
lines 66-67: Include the most recent IUCN reference about the species status.
Methodology:
lines 117-118: What is ProDelphinus NGO? Please, explain.
line 133: You mentioned 'fish species', but in fig. 5 there is the jumbo squid. Revise it here and along the text.
line 135: How do you know that 80% ethanol will not affect the isotopes and fatty acids resuls? You have to support it by references that tested it.
lines 137-139: Why are the sample sizes (liver vs. muscle) different? Please, explain. And about the sample sizes of other fish and squid?
line 188: The reference Kim et al. 2012 is not in the list. Why did you choose these values as discrimination factors?
Results:
lines 211-230: It is a long paragraph. Please, reorganize it.
Discussion:
lines 259-264: You only repeat the results here. Exclude this part from the text.
lines 266-267: It is a conclusion, and not a discussion.
lines 289-292: 13C is not a good tracer for trophic position. So, soften these phrases. Moreover, nothins is discussed about 15N, the most important stable isotope to evaluate trophic position. Discuss it here.
lines 301-302 and lines 327-388: Repeat the same information. Reorganize the text.
lines 326-336: You only repeat information from previous paragraphs. This subitem is not necessary in the ms.
lines 337-350: It sounds like a 'conclusion', but the reader will expect much more here. You have to show the real applicability of your results. Be more especific in what way your data will improve the local fishery management. Suggest some actions that can be done now, for instance. Describe the action (monitoring, regular sampling, etc), people involved (fishers, researchers, local government, etc) and expected results (delimiting or regulate fishing areas, track the sharks origin, etc).

Reviewer 2 ·

Basic reporting

no comment

Experimental design

Interestingly, the authors were cautious regarding the influences of lipids and urea on elasmobranchs, however they did attest that the samples were preserved in 80% ethanol, which is known to alter isotopic signatures, mainly for δ13C (Kelly et al. 2006; Kim & Koch, 2012). Therefore, the authors must consider this potential bias in the overall interpretation of results.

Validity of the findings

no comments

Additional comments

Here is my review of the ms 55838v1 by Segura-Cobeña et al. investiganting with stable isotope coupled with fatty acids data juvenile smooth hammerhead trophic niche of Humboldt Current System region of Peru. The informations are sound and improve the knowledge regarding nursery areas for sharks. Overall, I feel that the manuscript needs to be edited by authors in some questions about backgroud/context and hypothesis, sample treatment and interpretation of results that I describe below. For these reasons, I feel that a major revision and further round of peer review is necessary before this manuscript will be potentially suitable for publication.

General comments:
1- Some of the literature references should be updated by the authors. For example, the state of shark fin exporters in the world presented in the ms was from 15 years ago (line 64-65). Authors should present recent literature about this (last 5 years). Dent & Clark (2015) report regarding the state of the global market for shark products would be helpfull for the authors.

2- The ms lack of relevant results that sustain the hypothesis proposed by the authors. It was hypothesized that each nursery area is influenced by different water masses and, therefore, characterized by different environmental conditions, but the authors just presented that Máncora is located in the Tropical East Pacific Marine Province (TEP-MP), while San José and Salaverry are located in Warm Temperate Southeastern Pacific Marina Province (WTSP-MP). It was highlighted that WTSP-MP areas are characterized by high productivity driven by the Humboldt Current System (HCS) and high fish biomass and diversity, but no indication of possible differentes between WTSP-MP areas was presented to sustain the hypothesis. Additionally, it would be interesting the authors provide a short characterization of both water masses.

3- Regarding raw data, it seems the authors did not translated the acronym “M” that means “Mujer” to “F” that means “Female”, as well “H” that means “Hombre” to “M” that means “Male”. The authors have sex information for Máncora, but for San José and Salaverry. It seems there is no variation between sex within the same sampling area, but there is little variation between sex isotopes among the WTSP-MP areas. I took the liberty of calculating the δ13C and δ15N averages for each tissue and the isotopic variations cannot be neglected, especially that of δ13C in the liver ~ 2 ‰ between females. The muscle (San José: δ13Cfemale = -15.6, δ13Cmale = -15.7, δ15Nfemale = 16.0 and δ15Nmale = 15.8 ‰; Salaverry: δ13Cfemale = -16.4, δ13Cmale = -16.4, δ15Nfemale = 15.6 and δ15Nmale = 15.9 ‰) and liver (San José: δ13Cfemale = -17.0, δ13Cmale = -17.6, δ15Nfemale = 14.5 and δ15Nmale = 14.4 ‰; Salaverry: δ13Cfemale = -19.0, δ13Cmale = -18.5, δ15Nfemale = 14.2 and δ15Nmale = 14.2 ‰) data must be analyzed, interpreted and presented by the authors in the ms.

5- Discussion:
5.1. Characterization of isotopic niche variation variation among nursery areas - The authors refer to some previous studies in the literature, but do not provide clear evidence specific to this system to define the ENSO cycles niether the Oxygen Minimum Zone (OMZ) related to HCS for these nursery areas. Do any of these previous studies clearly identify them and/or present value ranges?

5.2. Characterization of FA profiles variation among nursery areas - The authors refer to some previous studies in the literature, but do not provide clear evidence specific to this system to define the FA for these three nursery areas. Do any of these previous studies clearly identify the FA for HCS and coastal upwelling?

5.3. Characterization of oceanic vs. coastal waters - The authors do not refer to any previous studies in the literature, and do not provide clear evidence specific to this system to define the δ13C end members for these two potential carbon sources. Do any previous studies clearly identify end members signals for oceanic and coastal waters inputs for this region?

Line-by-Line Comments:
Line 66 to 69: The IUCN Red List reference is also outdated. The doi took me to an actual IUCN Red List webpage that should be cited as "Rigby, C.L., Barreto, R., Carlson, J., Fernando, D., Fordham, S., Herman, K., Jabado, R.W., Liu, K.M., Marshall, A., Pacoureau, N., Romanov, E., Sherley, R.B. & Winker, H. 2019. Sphyrna zygaena. The IUCN Red List of Threatened Species 2019: e.T39388A2921825."

Line 74: Start a new paragraph in: “In the context of nature conservation,...”.

Line 124-125: Informations regarding El-Niño Southern Oscillation (ENSO) were not presented in this ms. The authors argue temperature and chlorophyll-a were stable despite seasonal variability related to ENSO. This data and absence of correlation should be presented in text and/or Supplementary Material.

Line 133-134: Were the sympatric species captured in the same period than Smooth Hammerhead?

Line 134: Change “that are captured” to “that are usually captured”.

Line 211: “The highest mean δ13C values” - The authors must specify if it is mean ± standard deviation (SD), mean ± standard error (SE) or mean ± confidence interval (CI).

Line 227-230: “These differences were consistent between results based on muscle and liver tissues. Sympatric fish species captured in all nursery areas were characterized by carbon signatures consistently enriched in 13C compared to carbon signatures of Smooth Hammerhead juveniles from the same areas (Fig 5)”. - Why authors plotted only liver tissue isotopic signatures of 13C and 15N of Smooth Hammerhead with muscle tissue of sympatric species?

Lines 237-243: I suggest authors to include Shannon diversity (H’) values and statistics informations in Table 3.

Lines 246-249: I suggest authors to include a table with permanova informations.

Tables 1 and 2 should be joined in a single table.

Title of Table 3: Authors should consider revision of Table 3 informations. Acronyms are not compatible with the acronyms shown in the table.
Legend of Table 3: “(two-way ANDEVA, P < 0.05)” - What is ANDEVA? I think authors meant ANOVA.

References:
Dent, F. & Clarke, S. (2015) State of the global market for shark products.
FAO Fisheries and Aquaculture Technical Paper No. 590. Rome, FAO. 187 pp .

Kelly, B., Dempson, J.B. and Power, M. (2006) The effects of preservation on fish tissue stable isotope signatures. Journal of Fish Biology, 69: 1595-1611. https://doi.org/10.1111/j.1095-8649.2006.01226.x

Kim, S.L., Koch, P.L. (2012) Methods to collect, preserve, and prepare elasmobranch tissues for stable isotope analysis. Environmental Biology of Fishes, 95: 53–63. https://doi.org/10.1007/s10641-011-9860-9

---

## Round 0.2 · accepted · Accept

The two reviewers agree that the suggestions have all been corrected. Only reviewer 2 makes a comment to the authors. Therefore, considering the revisions made in this version, I consider the manuscript accepted for publication.

Reviewer 1 ·

Basic reporting

All suggestions did in the first version of the manuscript were considered by the authors in this version or properly justified. This new version meets all the criteria of PeerJ and can be accept in the present form.

Experimental design

All suggestions did in the first version of the manuscript were considered by the authors in this version or properly justified. This new version meets all the criteria of PeerJ and can be accept in the present form.

Validity of the findings

All suggestions did in the first version of the manuscript were considered by the authors in this version or properly justified. This new version meets all the criteria of PeerJ and can be accept in the present form.

Additional comments

All suggestions did in the first version of the manuscript were considered by the authors in this version or properly justified. This new version meets all the criteria of PeerJ and can be accept in the present form.

Reviewer 2 ·

Basic reporting

no comment

Experimental design

no comment

Validity of the findings

no comment

Additional comments

The ms. 55838 by Segura-Cobeña et al. combines stable isotopes and fatty acids to address the trophic niches of smooth hummerhead among three nursery areas. I would like to congratulate the authors for the work done and for the consideration they had for my earlier comments. As a result, they largely improved the quality of the ms., notably as they succeed to efficiently support their hypothesys regarding spatial variation among HCS provinces and, as well, efficiently discussion their results, with clear characterization of isotopic niche variation among nursery areas and oceanic vs. coastal waters.

Minor comments:
Line 134: Acronyms must be specified: National Government Organization (NGO)
Lines 378 - 381: “We still found the results of muscle tissue analyses worth presenting as the fatty acid profiles that we found in liver tissues were similar to profiles found in muscle tissue found in the same individuals and, as expected, the concentrations in liver tissue were higher” - The authors use “found” repetedly in the same sentence. Please rewrite it.

Annotated reviews are not available for download in order to protect the identity of reviewers who chose to remain anonymous.